# Tuberculosis Infection in People Living with Human Immunodeficiency Virus: Challenges and Solutions

**DOI:** 10.3390/v16081295

**Published:** 2024-08-14

**Authors:** Ghassan Ilaiwy, Tania A. Thomas

**Affiliations:** Division of Infectious Diseases and International Health, Department of Medicine, University of Virginia, Charlottesville, VA 22903, USA

**Keywords:** tuberculosis infection, tuberculosis prevention, human immunodeficiency virus

The findings by Pipitò et al. highlight the challenges facing clinicians managing tuberculosis infection (TBI) among people living with human immunodeficiency virus (PLHIV), including the imperfect sensitivity of available tests and the slow adoption of shorter regimens of TBI treatment [1,2]. After excluding active tuberculosis (TB) disease, guidelines from the World Health Organization (WHO) recommend evaluating all PLHIV for TBI using either a tuberculin skin test (TST) or interferon-gamma release assay (IGRA), followed by TBI treatment with one of four recommended regimens or two alternative regimens for those who test positive or for all where testing is not available [3].

Regarding the first major challenge, a gold-standard test for TBI has yet to be identified, and both IGRA and TST have a suboptimal sensitivity which is further reduced in PLHIV [3,4]. IGRA outperformed TST in a systemic review by Chen et al., but its sensitivity of 64% was still suboptimal for detecting TBI among PLHIV [5]. Furthermore, the degree of immunosuppression further diminished the sensitivity of both tests with declining sensitivity among PLHIV with CD < 200 cell/μL that is more prominent for IGRA in low-income countries as reported by Cattamanchi et al. in their systematic review [6]. These diagnostic limitations call for a cautious interpretation of the TST and IGRA results, which should take into consideration the epidemiological background of PLHIV and the underlying TBI prevalence which can vary from 22.4% in the WHO Africa region to a low of 13.7% and 11% in the WHO Europe and Americas regions, respectively [7]. Therefore, individuals with PLHIV who were born in Africa have lower negative predictive values for TBI with a negative IGRA result than those with PLHIV who were born in Europe or the Americas. This would explain the lower prevalence of TBI highlighted by Pipitò et al. in the discussion of their findings, with only 7 out of 98 (7.14%) PLHIV who were born in Africa testing positive for TBI using IGRA [2].

The other major challenge highlighted is the attrition of PLHIV along the TBI care cascade even in a well-resourced setting. Pipitò et al. reported [5] the following outcomes of 38 PLHIV with positive IGRA: 7 (18.42%) lost to follow up before ascertaining their active TB status, 2 (5.26%) deferred treatment, 3 (7.89%) discontinued TBI treatment after initiation, 5 (13.16%) had ongoing treatment, and only 18 (47.46%) completed TBI treatment at the time of analysis. Despite guidelines in Italy incorporating three regimens for TBI: 4 months of once-daily rifampicin (4R), 3 months of daily isoniazid plus rifampin (3HR), or isoniazid daily for 6 or 9 months (6H or 9H) [8], all but one participant received either 6H or 9H. TBI treatment with 6H or 9H have been shown to have similar efficacy to shorter regimens such as 4R or 3HR but with a lower likelihood of completion in several studies in a variety of settings [9]. In one example, Menzies et al. found 4R to be non-inferior to 9H for TBI treatment in nine countries across the TB-prevalence spectrum, with higher completion rates and fewer adverse events in the 4R arm [10]. WHO guidelines also endorsed 3 months of weekly isoniazid and rifapentine and 1 month of daily isoniazid and rifapentine with their use so far hindered by the relative sparsity of data on use in children (especially younger than 2 years) and drug–drug interactions, especially in PLHIV [3].

Newer HIV regimens incorporating first-line Integrase Strand Transfer Inhibitor such as dolutegravir can help navigate drug–drug interactions with rifampicin and rifapentine, with emerging studies showing adequate dolutegravir exposure with twice daily dosing in the presence of rifampicin [11,12] and once daily dosing in the presence of rifapentine when used for TBI treatment [13]. Therefore, the use of dolutegravir-based first-line HIV regimens would further allow the use of shorter regimens for TBI treatments, which in turn would facilitate adherence among PLHIV and reduce the attrition frequently encountered in this setting. This approach was not widely considered in the data reported by Pipitò et al., with only one participant having a change in their HIV regimen. One can hypothesize that a change in regimen is less desirable among PLHIV on stable HIV regimens with excellent virologic control. However, this approach can be particularly helpful for people diagnosed with HIV and active TB at the same time.

TB has consistently ranked as one of the most lethal infectious diseases worldwide, claiming 1.3 lives in 2022 [14]. Starting in the 1980s, PLHIV have experienced a substantial portion of this burden as they are at least ten times more likely to progress or re-activate to active TB disease compared to the general population [15,16]. Tailored strategies to enhance the detection of TBI among PLHIV and improve their retention in care are crucial to lowering their risk of developing active TB disease and breaking the associated chains of transmission [3]. We join Pipitò and colleagues in their call for further innovation in TBI diagnostics to reduce the burden of TB and highlight the need for increased recognition of available shorter TBI treatment regimens to help retain more PLHIV in TBI care. Furthermore, an TBI test that identifies PLHIV who are at the highest risk for progression, while reflecting the current trend of improving HIV control and declining overall TB prevalence in the community, would go a long way in prioritizing available resources to improve TBI treatment outcomes.

## Data Availability

No new data were created or analyzed in this study. Data sharing is not applicable to this article.

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
