# Peer review of "Tuberculosis Infection in People Living with Human Immunodeficiency Virus: Challenges and Solutions"

_viruses, 2024, doi:10.3390/v16081295_

Round 1

Reviewer 1 Report

Comments and Suggestions for Authors

The author's surname Pipitò should be written with the accent on the final o.

Author Response

We thank the reviewer for their helpful comments and suggestions. Here'e a point-by-point response:

- The author's surname Pipitò should be written with the accent on the final o.

Response: Thank you very much for the suggestion, this was corrected as suggested throughout the manuscript (5 occurrences, lines 12, 33, 37, 61 and 71)

Reviewer 2 Report

Comments and Suggestions for Authors

Please check that all abbreviations are needed and that they are defined just once.

Page 2, line 65 (ref 14) is missing a word.

Some mention of PCR assays for TB would help.

Author Response

We thank the reviewer for their helpful comments and suggestions. Here'e a point-by-point response:

Reviewer 2 Comments:

- Please check that all abbreviations are needed and that they are defined just once.

Response: Thank you for the suggestions, removed abbreviations for 3HP, 1HP and INSTI as they only occurred once each in the text. Also made sure that all other abbreviations are defined at their first use and used consistently thereafter.

- Page 2, line 65 (ref 14) is missing a word.

Response: Thank you for drawing our attention to this, we removed the extra “the” in the sentence which now reads:
“TB has consistently ranked as one of the most lethal infectious diseases worldwide, claiming 1.3 lives in 2022”, lines 65-66

- Some mention of PCR assays for TB would help.

Response: PCR assays are used for the diagnosis of active TB disease as recommended by the World Health Organisation Guidelines (ref#3) and not merely TB infection. Therefore we respectfully believe that their mention does not fit well within the topic discussed in this manuscript.

Reviewer 3 Report

Comments and Suggestions for Authors

This opinion piece by Ilaiwy and Thomas highlights critical issues in managing latent tuberculosis infection (LTBI) among people living with HIV (PLHIV). This piece is timely and important as it calls for innovations in diagnostics and treatment approaches to address these ongoing challenges.

I have provided two small comments.

1. The manuscript uses the term "LTBI" throughout. It is worth noting that the World Health Organization (WHO) has recently updated its terminology, replacing "Latent Tuberculosis Infection (LTBI)" with "Tuberculosis Infection (TBI)." While some literature and guidelines still use the term LTBI, adopting the term TBI in this manuscript would align with the latest WHO recommendations and ensure consistency with current global health terminology.

2. There is a minor typographical error in the following sentence: "After excluding active tuberculosis (TB) disease, guidelines from the World Health Organization (WHO) recommend evaluating all PLHIV for LTBI using either a tuberculin skin test (TST) or interferon-gamma release assay (IGRA) followed by LTBI treatment with one of 4 recommended regimens or 2 alternative regimens for those who test positive or for all where testing is not unavailable." (Lines 14-18) The phrase "is not unavailable" should be corrected to "is not available" to accurately convey the intended meaning.

Author Response

We thank the reviewer for their helpful comments and suggestions. Here'e a point-by-point response:

Reviewer 3 Comments:

  1. The manuscript uses the term "LTBI" throughout. It is worth noting that the World Health Organization (WHO) has recently updated its terminology, replacing "Latent Tuberculosis Infection (LTBI)" with "Tuberculosis Infection (TBI)." While some literature and guidelines still use the term LTBI, adopting the term TBI in this manuscript would align with the latest WHO recommendations and ensure consistency with current global health terminology.

Response: Thank you for these suggestions, references to Latent TB infection were changed to TB infection and the used acronym changed to TBI throughout the manuscript (24 occurrences for the acronym).

  1. There is a minor typographical error in the following sentence: "After excluding active tuberculosis (TB) disease, guidelines from the World Health Organization (WHO) recommend evaluating all PLHIV for LTBI using either a tuberculin skin test (TST) or interferon-gamma release assay (IGRA) followed by LTBI treatment with one of 4 recommended regimens or 2 alternative regimens for those who test positive or for all where testing is not unavailable." (Lines 14-18) The phrase "is not unavailable" should be corrected to "is not available" to accurately convey the intended meaning.

Response: Thank you very much for the correction, this was corrected as suggested to now read: “(…)LTBI treatment with one of 4 recommended regimens or 2 alternative regimens for those who test positive or for all where testing is not available”, lines 18-19.